# Analyzing Similarity Metrics for Data Selection for Language Model Pretraining

**Dylan Sam**[*]
Carnegie Mellon University

**Ayan Chakrabarti**
Google Research

**Afshin Rostamizadeh**
Google Research

**Srikumar Ramalingam**
Google Research

**Gui Citovsky**
Google Research

**Sanjiv Kumar**
Google Research

## Abstract

Measuring similarity between training examples is critical for curating high-quality and diverse pretraining datasets for language models. However, similarity is typically computed with a generic off-the-shelf embedding model that has been trained for tasks such as retrieval. Whether these embedding-based similarity metrics are well-suited for pretraining data selection remains largely unexplored. In this paper, we propose a new framework to assess the suitability of a similarity metric specifically for data curation in language model pretraining applications. Our framework's first evaluation criterion captures how well distances reflect generalization in pretraining loss between different training examples. Next, we use each embedding model to guide a standard diversity-based data curation algorithm and measure its utility by pretraining a language model on the selected data and evaluating downstream task performance. Finally, we evaluate the capabilities of embeddings to distinguish between examples from different data sources. With these evaluations, we demonstrate that standard off-the-shelf embedding models are not well-suited for the pretraining data curation setting, underperforming even remarkably simple embeddings that are extracted from models trained on the same pretraining corpus. Our experiments are performed on the Pile, for pretraining a 1.7B parameter language model on 200B tokens. We believe our analysis and evaluation framework serves as a foundation for the future design of embeddings that specifically reason about similarity in pretraining datasets.

## 1 Introduction

The recent success of language models [Brown et al., 2020, Chowdhery et al., 2023] is in no small part due to pretraining on large and diverse text corpora scraped from a variety of sources [Raffel et al., 2020, Gao et al., 2020, Penedo et al., 2024]. Researchers have explored a variety of approaches to assemble effective pretraining sets, typically by selecting a high-quality and diverse subset from a larger corpus of examples scraped from multiple data sources. These data curation approaches have shown promising results by improving example quality and reducing redundancy in pretraining sets. Many of these methods use notions of *similarity* between examples. Similarity of an example to text from known high-quality sources (such as Wikipedia) has been used as a proxy for the quality of that example [Gunasekar et al., 2023, Penedo et al., 2024]. Meanwhile, methods focused on diversification [Abbas et al., 2023, Tirumala et al., 2023] make direct use of similarity metrics to identify and remove redundant examples.

In this context, similarity between training examples has often been measured in terms of distances in an embedding space. Many approaches have typically used generic embeddings for this purpose

---

[*]Work done while interning at Google Research. Correspondence to `dylansam@andrew.cmu.edu`.

39th Conference on Neural Information Processing Systems (NeurIPS 2025).

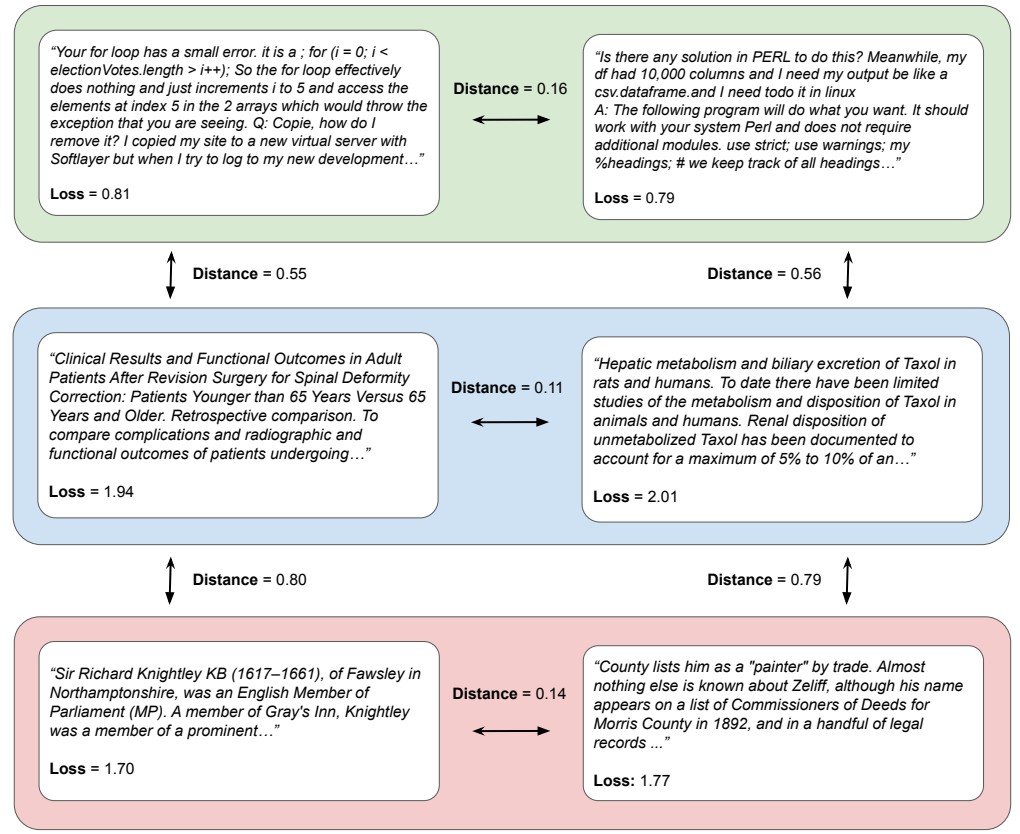

Figure 1: A visualization of the correlation between pretraining loss and embedding distance. Each row shows a pair of examples close in embedding space (from the same K-means cluster), with examples in different rows being far from each other (from different clusters). We find that close pairs of examples tend to have similar pretraining losses, while there is a greater variation in losses across clusters. Close example pairs are "thematically" similar but have different content. These results are from averaged embeddings from the final layer of a small decoder-only language model.

[Chang and Jia, 2023, Vo et al.]—off-the-shelf embedding models that have been trained for tasks such as semantic retrieval or mask-based reconstruction. However, whether these embeddings are optimal for reasoning about similarity between pretraining examples and whether there is any benefit to using more sophisticated and computationally expensive models remains an open question. Furthermore, evaluating such embeddings in pretraining applications is fundamentally challenging, as the sheer scale of pretraining data makes many experiments intractable. Such challenges necessitate specialized evaluations — new criteria for assessing the utility of embeddings in pretraining applications.

In this paper, we propose a novel evaluation framework to assess the suitability of an embedding model for curating pretraining data for language models. Our goal is to establish a **new standard** for evaluating similarity metrics in the context of language model pretraining. We begin by asking: what should a similarity metric ideally capture to be useful in this setting? Unlike prior work that evaluates embeddings for downstream tasks like retrieval or classification, we argue that embeddings used for data curation must reflect training dynamics—such as generalization behavior or corpus redundancy—that are unique to the pretraining setting.

Our first evaluation criterion measures how well distances in the embedding space correlate with generalization behavior during pretraining—specifically, whether nearby examples in embedding space exhibit similar pretraining loss under the same model state (see Fig. 1). This captures their utility of similarity metrics in loss-based data selection strategies [Jiang et al., 2023] or for selecting examples similar to high-quality references [Penedo et al., 2024]. Next, we evaluate the embedding model's utility in guiding a simple diversity-based curation method (similar to those employed in [Tirumala et al., 2023]) and assess its effectiveness by pretraining a model on the curated subset and measuring downstream task performance. Finally, we also measure whether the embedding space can distinguish between examples from different data sources. While not directly used in data selection,

this serves as a useful proxy for embedding quality under the assumption that these sources were curated by human expertise and reflect meaningful human-curated structure.

We conduct all of our experiments using the Pile [Gao et al., 2020] as our data corpus, and in the context of pretraining a 1.7B parameter decoder-only language model with a UL2 objective [Tay et al., 2022] on 200B tokens. We evaluate a number of different off-the-shelf embedding models—from representations of generic language models (e.g., trained with mask-based reconstruction or mixtures of pretraining objectives) to embeddings specifically trained for retrieval or semantic equivalence. We compare these to two specialized approaches that are derived from smaller, proxy models of the downstream model being trained on the same pretraining set: (1) computing the average token embedding at the final hidden state, and (2) a significantly more computationally efficient and simple average of the token-embeddings from the input layer (i.e., requiring **no forward passes**).

Despite their strong performance on standard semantic similarity and retrieval benchmarks, we find that off-the-shelf embedding models underperform in the context of pretraining data curation, suggesting that existing benchmarks may not reflect the inductive biases most relevant to language model pretraining. Surprisingly, even this simple method that averages input token embeddings—requiring no forward pass—matches or outperforms these more computationally intensive models. Adding a forward pass in this specialized model yields further improvements. These results strongly suggest that embedding models for reasoning about similarity between pretraining examples should be specialized to the data distribution at hand, and that properties that make embeddings suitable for retrieval and semantic matching may not transfer to the pretraining setting. More broadly, our work establishes a standardized framework to evaluate and highlight failures of current off-the-shelf models, facilitating the design of new models tailored for data curation in language model pretraining.

## 2 Related Work

**Data Curation.** Many works have studied the problem of selecting high-quality and informative subsets from a larger corpus for efficient and effective language pretraining [Albalak et al., 2024]. Indeed, works have shown that data curation for pretraining (both for language and vision-language models) improves neural scaling laws [Sorscher et al., 2022], and that curation techniques should be aware of the particular compute budget [Goyal et al., 2024].

One family of works approaches data curation with the goal of retaining only high quality examples. The simplest approaches in this family look at heuristics that can filter out noisy or extremely low-quality instances, such as documents containing less than 5 sentences or those with extremely short lines of text [Raffel et al., 2020, Xue, 2020]. Beyond these heuristic approaches, a line of work focuses on extracting high-quality examples based on similarity to known high-quality sources, such as Wikipedia or textbooks [Gunasekar et al., 2023, Penedo et al., 2024]. Other works look at creating targeted sets—defining quality in terms of their relevance for particular downstream tasks [Xie et al., 2023], with some creating these by adaptively updating weights over domain mixtures [Xie et al., 2024, Jiang et al., 2024]. The other family of data curation approaches focus on pruning datasets to ensure a notion of *diversity* and reducing redundancies in pretraining examples [Abbas et al., 2023, 2024, Tirumala et al., 2023]. While some of these works [Abbas et al., 2024, Tirumala et al., 2023] compare different embedding models as part of brief ablations, these comparisons are in the narrow context of their specific diversification algorithms. Finally, it is worth mentioning that some works in data selection use the model being trained as part of the selection process, often through the use of influence functions [Garima et al., 2020, Xia et al., 2024, Engstrom et al., 2024]. However, these are typically only used in data-curation on small datasets or only during finetuning, and would be prohibitively expensive in pretraining.

**Text Embedding Models.** Another related line of work is the task of learning embedding models for text. Many various approaches to learn text embedding models, with objectives including mask-based reconstruction [Devlin, 2018, Liu et al., 2019], a combination of multiple different tasks [Cer, 2018], and contrastive learning-based approaches [Gao et al., 2021, Neelakantan et al., 2022, Izacard et al., 2022, Lee et al., 2024]. More recent work has studied extracting embeddings from standard decoder-only language models [BehnamGhader et al., 2024], even via prompting [Sam et al., 2025].

These embedding models have largely been studied in the context for classification or similarity measures [Gomaa et al., 2013, Agirre et al., 2013, 2016], and recent focus has been on improving performance on aggregate on large-scale benchmarks [Muennighoff et al., 2022] that are comprised of multiple tasks (e.g., retrieval, clustering, classification, ranking). Many models that achieve

strong performance on these large-scale benchmarks have benefitted from scaling [Jiang et al., 2023, Chowdhery et al., 2023], with the help of distilled information from autoregressive models [Lee et al., 2024]. However, this has conflicting incentives with their utility in the pretraining setting, as large text embedding models are impractical and too computationally expensive to run inference over the full pretraining corpus. To the best of our knowledge, our work provides the first study of various text embedding models for pretraining data curation.

# 3  An Evaluation Framework for Embeddings in Pretraining Data Curation

We now describe our new framework for analyzing text embedding models in terms of their suitability for reasoning about similarity among pretraining examples. As previously mentioned, similarity is used to (a) find examples that are similar to known "desirable" examples (e.g., examples of known high-quality or those representing downstream tasks of interest) and to (b) discover and remove redundancies in the pretraining corpus. Accordingly, we design standardized experiments that measure a text embedding model's performance towards these criteria.

## 3.1  Evaluating Correlation with Pretraining Loss

We begin by evaluating whether low distances in a model's embedding space correlate with similar values of difficulty, which we measure by intermediate losses during language model pretraining. Examples of positive correlations are visualized in Figure 1. To do so, we first use a balanced K-Means clustering algorithm in the embedding space to cluster all examples in the pretraining set for a target cluster size. Then, we look at the variance of loss values within each cluster. This within-cluster variance captures whether points with similar pretraining loss (or difficulty) are grouped together and are close in embedding space. Crucially, we note that the converse is not necessarily true; two very different examples can have the same loss because they happen to be equally difficult to learn. This asymmetry is an important reason we chose our cluster-based approach rather than alternatives such as pairwise correlation. We note that we use balanced clustering (allowing only some variation in the sizes of different clusters) to ensure that average within-cluster variance is comparable across clusterings from different embedding models. We repeat this process for multiple target cluster sizes for all embeddings.

When reporting our results, we measure **variance reduction**, or the ratio of the overall variance across *all examples* in the pretraining dataset to the variance in loss computed *within clusters* in the given embedding space. Formally, we define variance reduction as

$$V(C) = \frac{E_{x \sim D}[(\ell(x) - E[\ell(x)])^2]}{E_{C_i \sim \mathbb{C}}\left[E_{x \sim C_i}[(\ell(x) - E_{x \sim C_i}[\ell(X)])^2]\right]},$$

where $C = \{C_1, ..., C_m\}$ is a clustering or partition of all datapoints in the dataset $D$, and $\mathbb{C}$ is a uniform distribution over each disjoint set in the partition $C$. Random clustering achieves a variance reduction of 1, and larger values of variance reduction imply that points with more similar pretraining loss are clustered together.

A high value of variance reduction implies that similarity in the embedding space correlates well with pretraining loss. While an isolated pair of examples having similar pretraining loss may be entirely unrelated in quality and content, the fact that an embedding space that *consistently* brings together examples of similar loss values strongly suggests that these similar examples will behave similarly in terms of contributing to the quality of the language model (when used in pretraining).

Beyond finding examples of similar quality and utility, a high variance reduction score implies that the clustering above can serve as a particularly useful proxy for more dynamic and online data sampling strategies [Xie et al., 2024], reducing the number of required forward passes in strategies where datapoints are selected based on their current pretraining loss. This also has implications towards approaches that look to self-improve models through propagating labeled information to nearby examples [Wei et al., 2021, Cai et al., 2021, Pukdee et al., 2023], such as weak-to-strong generalization [Burns et al., 2023].

## 3.2  Diversification-based Pretraining Data Curation

Prior work has demonstrated that data curation schemes that encourage diversification, based on similarity in embedding spaces, leads to improved models when trained on the curated subsets [Abbas

et al., 2023, Tirumala et al., 2023]. While the diversification techniques tend to be sophisticated, our goal here is to evaluate the utility of a given embedding for diversification. Therefore, we use a simplified version of these approaches to (1) select a subset from a pretraining corpus, (2) pretrain a model on this curated subset, and then (3) report performance on a large set of downstream tasks.

We begin by clustering the larger pretraining corpus in the given embedding space; in contrast to Section 3.1, we do not use balanced clustering, which can lead to clusters with a wide variation in "diameters" (distances among points in the same cluster). Instead, since the goal here is diversification, we find clusters such that distances between all pairs of points in the same cluster are within a specified threshold $\epsilon$.

Given clusters of pretraining data, prior work often uses complex pruning or sampling schemes from large clusters. For simplicity, we simply take the point in each cluster that is closest to the cluster centroid (i.e., the average embedding of all points in the cluster), which likely represents the most representative example for that cluster. We note that prior work [Tirumala et al., 2023] emphasizes the importance of a deduplication step in language model pretraining, and our curation strategy implicitly performs this process when selecting only a single point from each cluster.

Given that we select a single point from each cluster, this implies that we need the number of clusters to be equal to or greater than the number of desired pretraining examples. In practice, we sweep a large grid of $\epsilon$ values and select the largest threshold that still produces a sufficient number of clusters. Another important point of note is that producing such a large number of clusters is computationally expensive. To be able to scale to such a large number of clusters, we use the reciprocal agglomerative clustering (RAC) [Sumengen et al., 2021] algorithm. This has the advantage of building a clustering from progressively larger values of $\epsilon$, which works well with our need for a grid of values of $\epsilon$.

Finally, we pretrain a language model on the selected subset of examples, and measure few-shot performance of the pretrained model on a diverse variety of downstream tasks.

### 3.3 Measuring Cluster Purity with Respect to Data Sources

Large pretraining datasets such as the Pile [Gao et al., 2020] are comprised of various distinct yet complementary hand-curated data sources (e.g., high quality sources such as Wikipedia, code-based data, data from medical domains). While there may be some similarities between datapoints from different domains, we believe that clusters should generally group points from the same domain together and separate instances from different domains. In fact, since embedding models are generally not trained with explicit knowledge of source labels or any domain metadata, the ability to group together data from the same source and distinguish those from different sources shows an embedding model's alignment with meaningful human-curated structure (or at least that of the dataset curators). This also serves as a useful proxy of embedding model capacity, as they should contain sufficient information to well-separate these different sources.

Letting $s \in \{0, \cdots, S\}$ represent the index of the source in some finite number of sources $S$, we compute the **cluster purity** of a set of clusters $C$ as

$$P(C) = \mathbb{E}_{C_i \sim \mathbb{C}} \left[ \frac{\max_s |C_i \cap D_s|}{|C_i|} \right],$$

where $D_s$ represents the subset of datapoints that are from source $s$. Intuitively, this represents the proportion of the maximum frequency source to the total cluster size. A cluster purity of 1 represents a clustering that completely separates examples from different domains.

## 4 Experiments

We demonstrate the utility of our framework in evaluating off-the-shelf embedding models in measuring pretraining similarity and compare them with simple ways to produce specialized embeddings.

### 4.1 Experiment Details

**Embedding Models**   For the off-the-shelf embedding models, we consider: (1) Universal Sentence Encoders (**USE**) [Cer, 2018], which are general-purpose text encoders trained on a combination of objectives, (2) **Gecko** [Lee et al., 2024], a retrieval-focused text embedding model trained via synthetic data distilled from LLMs, and (3) **BERT** [Devlin, 2018] embedding models. These encapsulate a variety of different training objectives and reflect common embedding model choices in the field.

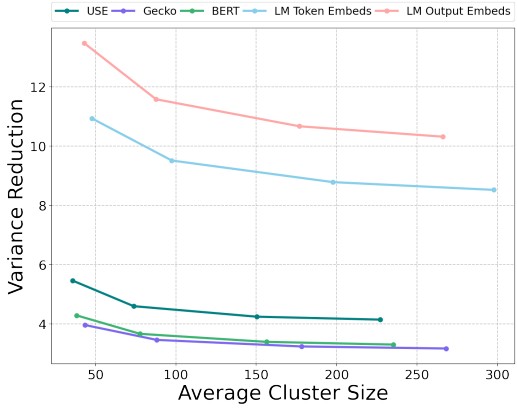 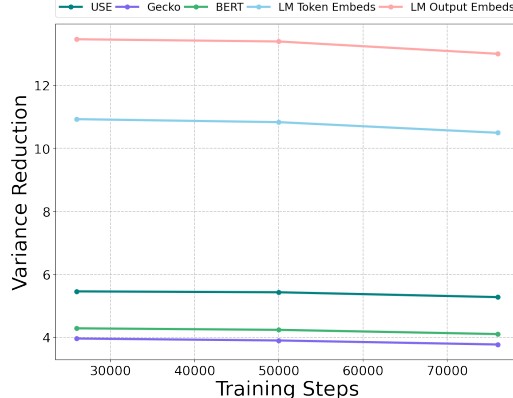

Figure 2: Variance reduction as we vary average cluster size. Larger values are better. Results are computed over 50 million sampled clusters from the Pile, where pretraining losses are computed after 26k gradient steps. **Specialized embeddings yield higher variance reduction than off-the-shelf models for all cluster sizes.**

Figure 3: Variance reduction as we increase the number of gradient steps in pretraining. Larger values are better. Results are computed over 50 million sampled clusters from the Pile with an average cluster size of 50. **Benefits in variance reduction remain consistent throughout pretraining**.

We also consider a few standard approaches to extract specialized embeddings that are extracted from a small version of the downstream language model we are training (e.g., a 200M parameter language model). The first and most simple approach is to extract an embedding by simply averaging the token embeddings matrix over all tokens in the input sequence, which we refer to as **LM Token Embeds**. This is extremely efficient as it only requires looking up the token embedding matrix and **does not involve any forward passes**. We note that this is also equivalent to a learned unigram model (i.e., ignoring all positional information). The second approach is to extract an embedding from the forward pass of the language model by averaging the activations over all tokens in the input sequence at the final hidden layer, which we refer to as **LM Output Embeds**.

Finally, in our diversity-based pretraining data curation experiments, we also add a comparison to a naive, random subset selection (**Baseline**). This evaluates how much benefit is observed from our standardized curation strategy with these various embedding models over standard baseline training.

**Clustering Details**    To make clustering with a large number of output clusters feasible and efficient at pretraining data scales, we perform dimensionality reduction using PCA on a large subset of the datapoints (∼500,000 examples) and extract the top 64 dimensions upon which to project. The projected embeddings are then normalized to have a unit L2 norm. We ablate on this choice for dimensionality reductions by comparing with random projections [Bingham and Mannila, 2001] as an alternative dimensionality reduction strategy; we find that random projections do not perform as well as PCA (Section 4.5). Other work [Tirumala et al., 2023] tackles this computational challenge by running clustering only with a much smaller subset of data, rather than performing dimensionality reduction and then clustering over all the data.

**Dataset and Pretraining Details**    For all of our experiments (e.g., pretraining data curation and predicting loss generalization), we use the Pile [Gao et al., 2020]. We pack together documents into a sequence of length 1280, with "[eod]" as delimiters between documents. For our pretraining experiments, we train a 1.7B parameter decoder-only language model with a UL2 objective [Tay et al., 2022]. We curated pretraining subsets from the Pile through the process outlined in Section 3.2. We use a selection budget of 200B tokens, or approximately 20% of the Pile. This corresponds to roughly 170 million clusters for each embedding model, with an average cluster size of 5 examples.

We pretrain with a learning rate of 0.001 with a linear decay and a batch size of 1024. For our tokenizer, we use sentencepiece with a vocabulary size of 256k tokens. For our pretraining evaluation, we consider the set of 23 downstream evaluation datasets from the work of Brown et al. [2020]. This involves a wide variety of 1-shot scoring tasks, as well as open-ended text generation tasks. More details about our evaluation sets are deferred to Appendix B.1. For the embeddings produced via

Table 1: Average downstream task performance of embedding models in the diversity-based curation schemes of a 200B token subset from the Pile, to pretrain a 1.7B parameter decoder-only language model. Bolding and italicizing denote the best and second-best performing methods on each task, respectively. Results are averaged over 3 pretraining runs, and average results are mean $\pm$ standard error. **Specialized embeddings** (i.e., extracted from a small version of the downstream language model) **and Gecko perform the best for diversity-based pretraining data curation**.

| Task | Baseline | USE | Gecko | BERT | LM Token Embeds | LM Output Embeds |
|---|---|---|---|---|---|---|
| ARC - Challenge | 32.4 | 33.0 | **33.7** | 32.7 | 32.5 | *33.5* |
| ARC - Easy | 63.8 | *65.1* | 64.2 | 65.1 | 64.9 | **65.5** |
| BoolQ | 56.5 | 59.2 | 58.3 | 60.1 | **62.9** | *61.9* |
| SuperGLUE - CB | 42.4 | 43.5 | *48.2* | 41.7 | 42.9 | **48.8** |
| SuperGLUE - Copa | 75.3 | 76.3 | *77.3* | 74.0 | 74.7 | **78.0** |
| HellaSwag | 55.0 | 56.7 | *57.1* | 56.5 | 56.8 | **57.5** |
| Multi RC | **57.7** | 55.7 | 55.9 | *56.3* | 56.0 | 55.2 |
| OpenBook QA | 46.3 | 46.2 | 46.3 | *46.6* | 45.9 | **46.7** |
| PiQA | 72.3 | 73.0 | 72.7 | *73.6* | 72.8 | **73.6** |
| Race H | 38.0 | 37.9 | 38.5 | *38.8* | **38.8** | 38.7 |
| Race M | 51.7 | 52.0 | **53.3** | *52.9* | 52.1 | 52.6 |
| ReCoRD | **85.0** | 84.4 | 84.6 | 84.7 | 84.5 | *84.9* |
| RTE | *54.5* | **54.9** | 51.5 | **54.9** | 51.6 | 52.8 |
| Story Cloze | 73.9 | 74.1 | *74.3* | 74.0 | 73.3 | **74.5** |
| WiC | **48.2** | 47.9 | 47.5 | 47.5 | *48.2* | 47.3 |
| Winograd | 74.0 | 76.0 | *77.1* | **77.3** | *77.1* | 76.6 |
| WinoGrande | 59.2 | 58.9 | 59.0 | *59.5* | **60.0** | 59.3 |
| WSC | 74.1 | 73.9 | 73.7 | *74.7* | 73.2 | **75.0** |
| Lambada | 21.4 | 29.9 | **40.1** | *34.7* | 31.4 | 33.8 |
| Natural Questions | 10.0 | 10.8 | 10.1 | 10.5 | *11.1* | **11.4** |
| SQuAD v2 | 51.1 | 54.3 | 51.7 | 51.8 | **58.6** | *55.4* |
| TriviaQA Wiki | 34.4 | *35.0* | 33.0 | 35.0 | 34.3 | **36.1** |
| Web Questions | 17.1 | 17.0 | 17.9 | 18.0 | *18.7* | **19.1** |
| **Average** | $51.9 \pm 0.1$ | $52.9 \pm 0.1$ | *53.3 ± 0.3* | $53.1 \pm 0.2$ | $53.1 \pm 0.3$ | **53.8 ± 0.1** |

LM Token Embeds and LM Output Embeds, we train a 200M parameter language model in the same fashion as above, where we also train on a total of 200B randomly selected tokens.

## 4.2 Pretraining Loss Correlation Results

To measure the ability of embedding models to reflect pretraining loss generalization and to cluster together datapoints with similar difficulty, we report their variance reduction as we vary (1) the number of gradient updates performed by the intermediate pretraining checkpoint and (2) as we vary the average cluster size in our K-Means clustering.

We observe that specialized embeddings (i.e., LM Token Embeds and LM Output Embeds) from a small version of the downstream trained model, *which is trained on the same pretraining data*, achieve a much larger reduction in loss variance when compared to off-the-shelf models. These specialized embeddings produce better clusters with datapoints with similar intermediate pretraining losses, across all cluster sizes (Figure 2) and for all intermediate pretraining checkpoints (Figure 3). This suggests that training models specifically on the data to perform data curation is preferable to using off-the-shelf embedding models for predicting pretraining loss generalization. Furthermore, we also generally observe that newer embedding models (e.g., Gecko) do not improve upon older alternatives (e.g., BERT and USE), supporting that embedding model improvements in other settings, such as retrieval *do not translate* to this application in the pretraining setting.

**Visualizing Within-Cluster Examples**   To better interpret these similarity metrics, we visualize pairs of datapoints from the clusters in Figure 1 to understand what types of examples are placed closely together and which have similar pretraining losses. We randomly sample 3 clusters produced by K-Means clusters with the smallest average cluster size, with the LM Output Embeds method to extract embeddings. We randomly sample 2 points from these 3 clusters and observe that samples

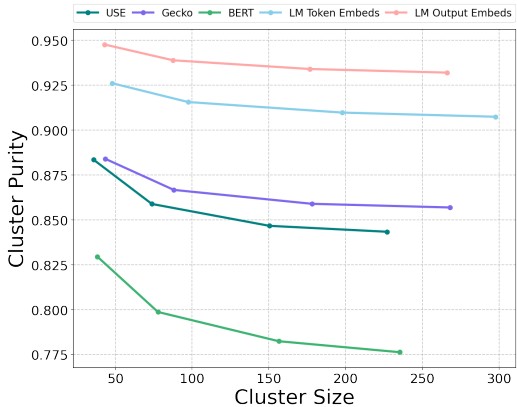

Figure 4: Comparison of the purity with respect to data source of K-Means clustering produced by various embedding models on the Pile, when averaged over 50 million clusters from the Pile. **Specialized embedding models have higher cluster purity scores.**

from the same cluster are thematically similar, although containing very different information (e.g., the last pair referring to two different government-related people, although from different times and different countries). We observe that pretraining losses are similar among within-cluster examples, and lower distances correspond to smaller differences in pretraining loss.

## 4.3    Diversification-based Pretraining Data Curation Results

We report the average downstream task performance of language models trained on pretraining data subsets produced by using various embedding models in the simple diversity-based data curation strategy in Table 1. We first observe that the standardized diversity-based data curation technique applied to all embedding models (both off-the-shelf and those based on small downstream trained language models) outperforms the naive random subset selection baseline. Secondly, we observe that the LM Output Embeds achieves the best performance when averaged across all tasks.

A notable result is that LM Token Embeds performs comparably to Gecko, while outperforming all other off-the-shelf embeddings. This is of immediate practical interest due to the simplicity of LM Token Embeds; they require no forward pass and, consequently, are extremely quick to compute in comparison to all other considered alternatives. Thus, LM Token Embeds serves as a viable alternative in settings that are compute-limited. This also suggests that even simple notions of similarity, even those that do not account for positional information, are sufficient for many pretraining data curation applications. Finally, they support that models specialized to this task (e.g., trained on the same dataset) often outperform the current, general-purpose, off-the-shelf embedding models. While this does include training an additional model, it is a single-time investment that is amortized across all embedding use cases and is easily offset by the significant improvements in data curation quality.

## 4.4    Cluster Purity Results

To better understand and interpret the similarity metrics defined by these embedding models, we measure the purity of clusters with respect to the underlying data source. We again remark that none of these embedding models have been trained with knowledge of the data source. Overall, we observe that most embedding models produce fairly pure clusters, where a majority of points come from the same underlying data source (Figure 4). This supports that these embeddings models are generally aligned with human judgment about differences between types of data.

We observe that the embeddings extracted from the small version of the trained downstream language model achieve the highest cluster purity. We also note that Gecko embeddings achieve the third-highest cluster purity, whereas it performs poorly on the task of predicting loss generalization through variance reduction. This suggests that an improved ability to predict pretraining loss generalization cannot simply be explained via producing clusters that are more pure with respect to underlying data sources. Future work could explore incorporating domain information or metadata into the embedding model's learning objective, as a form of weak supervision [Sam and Kolter, 2023].

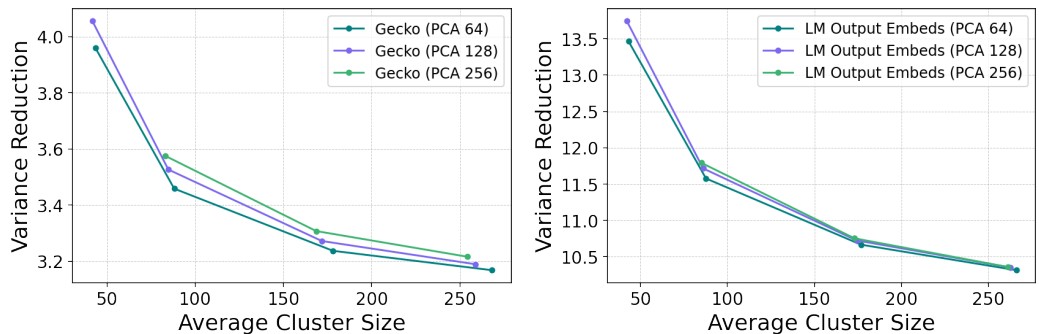

Figure 5: Ablation on the number of components in PCA for Gecko and LM Output Embeds. Results are averaged over 50 million sampled clusters from the Pile. **Using more components in PCA better clusters points with similar pretraining loss.**

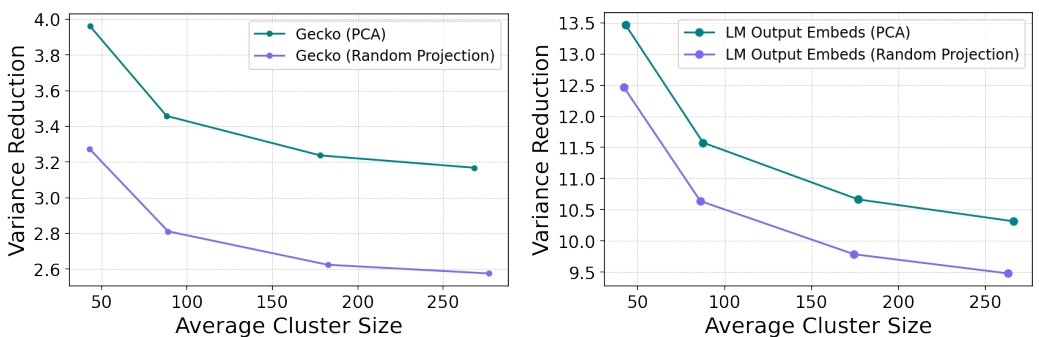

Figure 6: Ablation comparing the use of PCA or Random projections for dimensionality reduction in Gecko and LM Output Embeds. Results are averaged over 50 million sampled clusters from the Pile. **Dimensionality reduction via PCA performs better than via Random Projections.**

## 4.5 Ablations

We present ablations on components of our evaluation framework — specifically in how we have chosen to perform dimensionality reduction (i.e., technique and resulting size), which is used for the embedding models before running clustering. We present the ablations for Gecko and LM Output Embeds and defer results on others embeddings to Appendix A.

**Less Components in PCA Does Not Significantly Hurt Performance.** The dimensionality of the embeddings used in clustering often must be low for pretraining scales. Here, we run an ablation studying the role of dimensionality (i.e., the number of components in PCA onto which the embeddings are projected) in the variance reduction through clustering (Figure 5). We remark that scaling clustering to accommodate smaller average cluster sizes (i.e., more cluster centers) is intractable for embeddings that have high dimensionality. Thus, we can only report results on various embeddings with a large number (i.e., 256) of components in PCA with larger average cluster sizes. We observe the trend across all embeddings that a higher dimension and larger number of PCA components improves the embedding models' ability to cluster points by pretraining loss (see other embedding model results in Appendix A.1).

**Dimensionality Reduction with PCA Outperforms Random Projections.** Another common technique to perform dimensionality reduction (especially with respect to maintaining pairwise distances and similarities) is to use random projections. We experiment with using random projections in our loss clustering experiments and observe that it is outperformed by PCA in terms of variance reduction across all cluster sizes (Figure 6).

# 5 Discussion

We present a new evaluation framework for the understudied area of similarity metrics used in language model pretraining data curation. Using our framework, we show that off-the-shelf embeddings—despite their widespread use—often underperform even simple, specialized alternatives such as an average of token embeddings that requires no forward pass. This also suggests that practitioners should train their own embedding models, and it suffices to train them on a small fraction of the data (∼20%) and at a much smaller parameter scale. While it may seem intuitive that embeddings tailored to the pretraining task would outperform generic ones, this has not been systematically demonstrated and is certainly not yet standard practice. Our framework both surfaces this gap and offers a practical tool for guiding the design of embedding models optimized for data curation, which can lead to significant improvements in data efficiency.

While we believe that our results are fairly general, future work could extend our findings to other pretraining corpora. Beyond evaluation, our findings have broader implications for pretraining workflows—for example, in selecting task-specific finetuning data [Xia et al., 2024], or identifying synthetic examples that resemble real data [Meng et al., 2022, Sam et al., 2024], and even studying the impacts of scaling up embedding models in these tasks. Overall, our results underscore the importance of using task-aligned similarity metrics in pretraining, and our framework provides a foundation for future research in building and optimizing the design of embedding models tailored to this critical step of the language modeling pipeline.

**Limitations** One limitation of our work is that the empirical demonstration of our new evaluation framework is restricted to the Pile dataset. We believe this is fairly standard in the language model pretraining data curation literature to focus on a single pretraining corpus (e.g., as is done in Tirumala et al. [2023]), due to the computational costs of both data curation algorithms and pretraining.

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

# A  Additional Experimental Results

## A.1  Impact of Dimensionality Reduction on Variance Reduction

We now present the remaining results for other embedding models, when we ablate the number of components used in PCA for K-Means clustering, specifically when looking at the reduction in variance of pretraining loss of points within the same cluster. We observe that more components in PCA indeed help achieve higher variance reduction across all embedding models.

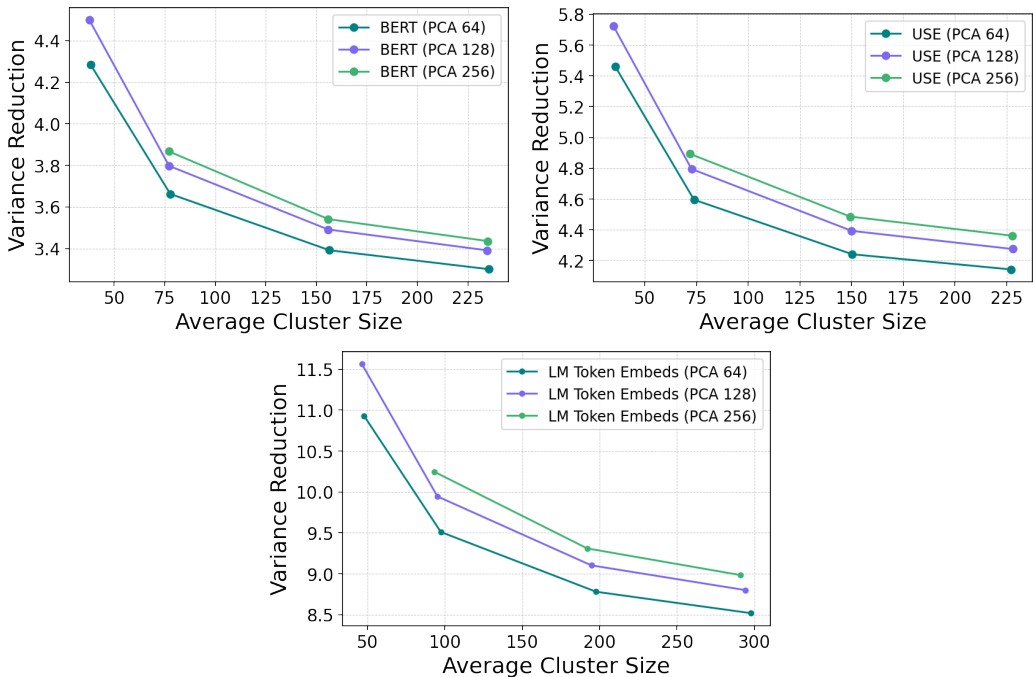

Figure 7: Ablation on the number of components in PCA for embeddings from BERT, USE, and LM Token Embeds. Results are averaged over 50 million sampled clusters from the Pile.

## A.2  Comparison of Random Projections to PCA for Dimensionality Reduction

We now present the remaining results for other embedding models, when we use Random Projections for dimensionality reduction instead of using PCA. We consistently observe that embeddings paired with PCA outperform those using Random Projections.

# B  Additional Experimental Details

## B.1  Additional Evaluation Details

We report results averaged over a large number of downstream tasks. These datasets largely follow two categories: scoring and decoding tasks. Scoring tasks primarily look at the output distribution of the model, while decoding tasks look at text generations from the language model. Scoring tasks are performed as 1-shot (i.e., giving one demonstration of format and answer), while decoding is performed zeroshot. For scoring tasks, we look at the standard top-1 accuracy. The list of scoring tasks is as follows:

- ARC Challenge: Easy and Challenge [Clark et al., 2018]
- BoolQ [Clark et al., 2019]
- SuperGLUE - CB and Copa [Wang et al., 2019]
- HellaSwag [Zellers et al., 2019]

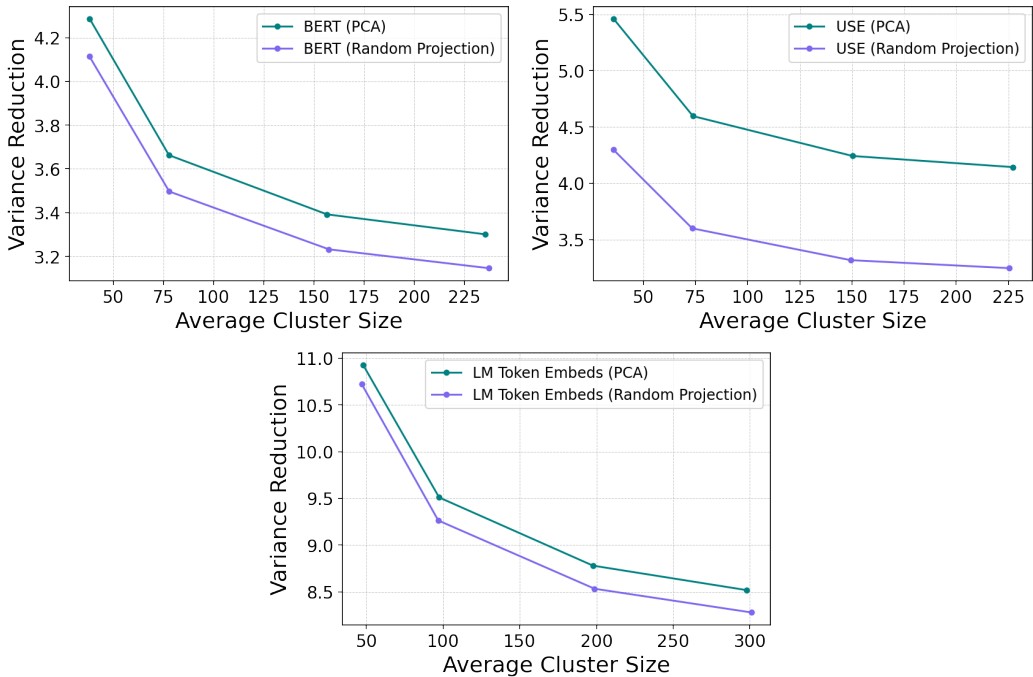

Figure 8: Ablation where we compare using Random Projections for dimensionality reduction with embeddings from BERT, USE, and LM Token Embeds. Results are averaged over 50 million sampled clusters from the Pile.

- MultiRC [Khashabi et al., 2018]
- OpenBookQA [Mihaylov et al., 2018]
- PIQA [Bisk et al., 2020]
- RACE-H and RACE-M [Lai et al., 2017]
- ReCoRD [Zhang et al., 2018]
- RTE [Dagan et al., 2010]
- Story Cloze [Mostafazadeh et al., 2016]
- WIC [Pilehvar and Camacho-Collados, 2018]
- Winograd [Levesque et al., 2012]
- WinoGrande [Sakaguchi et al., 2021]
- WSC [Levesque et al., 2012]

For decoding or text generation tasks, we evaluate the language model outputs with its F1 score. Decoder tasks are also evaluated as 1-shot. The list of decoding tasks is as follows:

- Lambada [Paperno et al., 2016]
- Natural Questions [Kwiatkowski et al., 2019]
- Squad v2 [Rajpurkar et al., 2018]
- Trivia QA Wiki [Joshi et al., 2017]
- Web Questions [Berant et al., 2013]

## B.2  Additional Embedding Model Details

The embedding models of BERT, Gecko and USE are trained with sequence lengths of 512, which we apply on the first 512 tokens of data from the Pile. For Gecko, we use the 110 million parameter model version, while for USE we use the 109 million parameter version. For BERT, we also use a

109 million parameter model. For the small language model that we train with the UL2 objective, we use one with approximately 200 million parameters.

Both the off-the-shelf BERT and USE embedding models have a dimensionality of 512. The Gecko embedding model has a dimensionality of 768. The small language model has a token embedding dimension of 512 and an hidden activation dimension of 512, which means that both LM Token Embeds and LM Output Embeds have 512 dimensions.

### B.3 Additional Hyperparameter Details

**Clustering**    For performing RAC clustering for our pretraining experiments, we use a value of $\epsilon$ as the particular diameter of clusters:

- USE: $\epsilon = 0.2$, which defines roughly 225 million clusters
- Gecko: $\epsilon = 0.2$, which defines roughly 220 million clusters
- BERT: $\epsilon = 0.001$, which defines roughly 175 million clusters
- LM Token Embeds: $\epsilon = 0.001$, which defines roughly 170 million clusters
- LM Output Embeds: $\epsilon = 0.03$, which defines roughly 180 million clusters

For our K-Means clustering, we perform clustering at 4 different levels of granularity in our variance reduction and cluster purity results. We create four sets of clusterings with an average cluster size of 25, 50, 100, 150, with a minimum cluster size of $\frac{1}{5}$ times the average cluster size, and a maximum cluster size of $5$ times the average cluster size. For both RAC and K-Means, we use the squared L2 (Euclidean) distance.

### B.4 Dimensionality Reduction Details

For running our dimensionality reduction via PCA, we compute the means and components on which to project on a subset of the data ($\sim$500,000 points). We first standardize the embeddings to have a mean of 0 and variance of 1 before running PCA. As previously mentioned, after projecting onto the desired number of principal components, we perform L2 normalization.

For our random projections, we use a sparse random projections onto values of $-\frac{\sqrt{n}}{\sqrt{64}}, 0, \frac{\sqrt{n}}{\sqrt{64}}$ with probabilities $\frac{1}{4}, \frac{1}{2}, \frac{1}{4}$ respectively (i.e., the default parameters via scikit-learn). We also L2 normalize the result of random projections.

### B.5 Compute Details

Pretraining experiments for our 1.7B parameter language models are run on 512 v5 TPUs, where each pretraining run takes approximately 3 days. Training our proxy 200M parameter model took less than 1 day on 64 v5 TPUs. Hierarchical clustering for pretraining requires approximately 1-2 days to run over the full pretraining corpus.

### B.6 Asset Licenses

The existing assets that we use have the following licenses:

- RAC Clustering: MIT license
- The Pile: CC BY 4.0
- Evaluation Datasets: MIT License

