# OpenReview forum: "Analyzing Similarity Metrics for Data Selection for Language Model Pretraining"
_NeurIPS.cc/2025/Conference — NeurIPS 2025 poster_

### Official Review · Reviewer_Q6cD · 2025-06-29

**Clarity:** 4
**Significance:** 2
**Originality:** 3
**Rating:** 4
**Confidence:** 3

**Summary:**

* This work focuses on in-depth evaluation of how text embedding based data curation can improve LLM pretraining.
* There are two overall concepts for data curation presented in this work:
  * When representation similarity is desired: finding examples that are similar to hard examples, or high quality examples.
  * When representation similarity is not desired: finding diverse examples to remove training data redundancies.
* The three part evaluation that this work presents involve: loss-correlation test, diversity-based curation test and cluster-purity test. These metrics are closely tied to pre-training objectives versus classical embedding model task evaluations.

**Questions:**

1. Recent embedding models such as e5-mistral and nv-embed support task prompts (clustering, semantic similarity etc.). Even more recent models such as BGE-EN-ICL support few-shot in-context learning ability. How could those abilities be leveraged to customize the representations for the various tasks (when the desired similarity should be based on problem difficulty vs. text quality vs. topic).
2. "LLM2Vec: Large Language Models Are Secretly Powerful Text Encoders" could be referenced to justify why LM Output Embeds in Table 1 works well.

**Ethical Concerns:**

["NO or VERY MINOR ethics concerns only"]

**Final Justification:**

I thank the authors for taking the time to give detailed responses to each question. I agree with the author's responses and there are no unresolved issues in particular. I stand by the original score of 4 and would reserve a score of 5 or 6 for if the work has demonstrated even higher downstream task performance and implemented the guidance in their response for loss-correlation and cluster-purity tests to show further improvements in downstream task performance or data efficiency.

**Limitations:**

Yes.

**Paper Formatting Concerns:**

None.

**Quality:**

3

**Strengths And Weaknesses:**

Strengths:
1. Experiments are rigorous, and methodology is sound. Especially the three tests, downstream task performance, and coverage of various possible embedding models and direct-from-LLM representation approaches.
2. Data curation is an important aspect of pretraining and rigorous studies such as this work is helpful for the field.
3. The framing of embedding models in-relation to pretraining objectives (instead of classic MTEB benchmark tasks) is original.

Weaknesses:
1. The empirical results (downstream task performance) are modest (within 2%).
2. Only diversity-based curation is ultimately linked to downstream task performance with results. There is no guidance or results presented on how loss-correlation and cluster-purity tests can be leveraged to improve downstream task performance.
3. Extending the evaluations beyond the single corpus and model size would make the study more rigorous, but I understand that it is very computationally expensive.

---

> ### Author Rebuttal · Authors · 2025-07-31
>
> We thank the reviewer for their positive and constructive feedback. We are glad you found our experiments "rigorous," our methodology "sound," and our framing "original" and "helpful for the field." We address your points below.
>
> > **"The empirical results (downstream task performance) are modest (within 2%)."**
>
> This is a fair observation. While a ~2% gain may seem modest, we would like to emphasize two points. First, this is an average improvement across **23 downstream tasks**, indicating a consistent and broad enhancement of the model's capabilities rather than a narrow gain on a single benchmark. Second, and perhaps more importantly, this performance gain can be viewed through the lens of **data efficiency**. Our curation method allows the model to reach a target performance level using significantly fewer pretraining tokens compared to random sampling, directly translating to substantial savings in compute and time. We will add this data efficiency perspective to our revision to better contextualize the impact of our results.
>
> > **"Only diversity-based curation is ultimately linked to downstream task performance with results. There is no guidance or results presented on how loss-correlation and cluster-purity tests can be leveraged to improve downstream task performance."**
>
> Loss correlation can be directly tied to target-specific sampling schemes (which we state in lines 149-151), but we are happy to expand upon this and provide more concrete guidance in a potential application. The variance reduction metric (Section 3.1) captures how well an embedding reflects training difficulty as measured by pretraining loss. Embeddings with high variance reduction can be used to group examples by difficulty, which directly informs a family of sampling schemes known to improve pretraining efficiency and performance—e.g., importance resampling or dynamic curriculum methods (as in Xie et al. 2024, Jiang et al. 2024).
>
> One such method is an online training strategy, where at each step, you can adaptively select the next cluster to train that has the largest pretraining loss. With embeddings that produce clusters with greater variance reduction, this means that one can run a single forward pass on the cluster centers (rather than all datapoints) and use those values as more accurate estimates of cluster loss.
>
> With cluster sizes of 50–200, this can yield 50–200× savings in selection overhead compared to per-example loss estimates. Our framework thus enables **scalable and efficient proxies for loss-aware data selection**.
>
> As for cluster purity, it serves as a useful and interpretable diagnostic: embeddings that align with human-curated data sources (e.g., Wikipedia vs. code) are more interpretable and trustworthy for **building specialized datasets or controlling data mixtures**. We envision this guiding domain-aware curation, where users seek to curate or balance datasets along interpretable axes. We will include both of these examples in our revision.
>
> > **"Recent embedding models such as e5-mistral and nv-embed support task prompts (clustering, semantic similarity etc.). Even more recent models such as BGE-EN-ICL support few-shot in-context learning ability. How could those abilities be leveraged to customize the representations for the various tasks (when the desired similarity should be based on problem difficulty vs. text quality vs. topic)."**
>
> That's an excellent point. Promptable and in-context embedding models offer exciting possibilities for tailoring similarity to task-specific notions—such as difficulty, redundancy, or style. Our evaluation framework would be an ideal tool for assessing how effectively these prompts can steer the embeddings to capture these different notions of similarity. We will add this discussion on embedding models that can leverage in-context learning as a promising future direction in our discussion.
>
> > **"LLM2Vec: Large Language Models Are Secretly Powerful Text Encoders" could be referenced to justify why LM Output Embeds in Table 1 works well."**
>
> Thanks for this pointer! We agree that LLM2Vec provides strong, complementary evidence for our finding that embeddings from decoder-only models can be highly effective. We will add this citation to our related work section in the revision.
>
> Thank you again for your detailed review. We hope these clarifications address your concerns and help reinforce the value of our contributions.

---

> > ### Comment · Reviewer_Q6cD · 2025-08-05
> >
> > Thank you for the detailed clarifications. I do not have any further issues or questions. I am maintaining my score of 4 since only one of the metric (diversity-based curation) is actually used end-to-end to improve downstream task performance. For this work, I reserve the score of 5 or 6 for 1) if the loss correlation metric guidance that you mention was also implemented to show improved downstream task performance and 2) the ICL or task prompted retrievers were studied in the submitted paper since they are relevant to this approach.
> > One suggestion for showing downstream task performance when an approach is meant more for data efficiency is to compare between A) the performance with the efficiently subsampled dataset with the applied approach and B) the performance with the original dataset randomly subsampled to the same size as the efficiently subsampled dataset.

---

> > > ### Author Response · Authors · 2025-08-07
> > >
> > > Thank you again for your review and continued engagement! We appreciate your positive recommendation of our work.
> > >
> > > ---
> > >
> > > We'd like to reiterate that the main contribution of our work is a framework and metric to evaluate different embedding models that can be done scalably and efficiently.
> > >
> > > Regarding (1), developing such an approach (such as the simple one that we suggested) for adaptive learning or curriculum learning is in itself an interesting and related open area of research, which is actively being studied. Also, we'd like to emphasize that evaluation metrics are crucial to be **computationally efficient**, and running demonstrations on adaptive pretraining runs are extremely computationally intensive (e.g., each pretraining run in our diversity-based pretraining analysis requires $20000 in Cloud TPU credits). With regards to cluster purity, we'd also like to point the reviewer to relevant work [3], which highlights the **benefits of constructing domains** from a single monolithic web corpora. This is certainly related to our notion of cluster purity and reflects how this is a useful proxy metric.
> > >
> > > Regarding promptable and in-context embedding models, we completely agree that this is a fascinating direction. Our framework is perfectly suited to systematically evaluate how effectively different task prompts or few-shot examples could steer embeddings to capture various notions of similarity (e.g., difficulty vs. topic). We have added this as a promising future direction in our discussion, thanks to your suggestion.
> > >
> > > [1] Wettig, et. al. Organize the Web: Constructing Domains Enhances Pre-Training Data Curation.
> > >
> > > ---
> > >
> > > Thank you again for all of your efforts in the review process!

---

### Official Review · Reviewer_VPjN · 2025-07-03

**Clarity:** 2
**Significance:** 2
**Originality:** 2
**Rating:** 3
**Confidence:** 3

**Summary:**

This paper proposes a framework for evaluating embedding models in the context of language model pretraining. The authors aim to set a new standard for how similarity metrics should be assessed for this use case. They start with the question: what should a similarity metric capture to be useful for curating pretraining data? Their main claim is that embeddings used for this purpose should reflect training dynamics, like generalization behavior and redundancy in the dataset, rather than downstream retrieval or classification performance. The paper introduces three evaluation criteria: (1) do distances in embedding space correlate with pretraining loss, (2) how well do embeddings support diversity-based selection methods, and (3) can embeddings distinguish between examples from different data sources. The experiments are run on the Pile with a 1.7B decoder-only model trained with a UL2 objective on 200B tokens. They compare standard off-the-shelf embeddings to simple embeddings taken from smaller models trained on the same corpus. These in-domain embeddings, even ones computed without a forward pass, perform better across most metrics.

**Questions:**

1. Can you provide a direct analysis of the relationship between embedding distance and pretraining loss similarity, rather than relying solely on cluster-based variance reduction? This would help clarify whether your method supports the claim that embedding distance reflects generalization behavior.

2. How do you address the potential circularity in using embeddings from models trained on the same data used in the evaluation? Would these embeddings generalize to new domains or corpora where no model has yet been trained?

3. Can you report more complete results for the variance reduction analysis, such as how it changes across different cluster sizes, seeds, or pretraining checkpoints? The current presentation makes it hard to assess the robustness of your findings.

4. Why did you choose balanced K-means for clustering? Have you tested how sensitive your results are to the clustering algorithm, and could this choice be inflating variance reduction?

**Ethical Concerns:**

["NO or VERY MINOR ethics concerns only"]

**Final Justification:**

Given the rebuttal, I still believe the paper should be rejected. While the authors clarify their choice of cluster-based variance over pairwise correlation, this still does not directly demonstrate that embedding distance captures the training dynamics they claim. The defense of balanced K-means addresses stability but not whether it inflates results, and the generalization concern remains; the proposed approach assumes retraining an embedding model for every new corpus, which is impractical in many settings. Overall, the evaluation is limited, and the framing overstates the generality and efficiency of the method.

**Limitations:**

1. Can you provide a direct analysis of the relationship between embedding distance and pretraining loss similarity, rather than relying solely on cluster-based variance reduction? This would help clarify whether your method supports the claim that embedding distance reflects generalization behavior.

2. How do you address the potential circularity in using embeddings from models trained on the same data used in the evaluation? Would these embeddings generalize to new domains or corpora where no model has yet been trained?

3. Can you report more complete results for the variance reduction analysis, such as how it changes across different cluster sizes, seeds, or pretraining checkpoints? The current presentation makes it hard to assess the robustness of your findings.

4. Why did you choose balanced K-means for clustering? Have you tested how sensitive your results are to the clustering algorithm, and could this choice be inflating variance reduction?

**Quality:**

2

**Strengths And Weaknesses:**

$\textbf{Strengths}$

- The paper addresses an interesting gap in the literature. People use similarity metrics to filter and select pretraining data. However, these metrics are rarely designed or evaluated with this specific setting in mind.

The authors propose a clear framework for assessing embeddings that is well-suited to the pretraining context.

- Some of the results are surprising. Very simple embeddings derived from the training corpus outperform more complex, pretrained models.

$\textbf{Weaknesses}$

- The strongest claims rely on the idea that low embedding distances mean similar pretraining loss, but the actual evaluation uses variance within clusters, not pairwise distance. This does not prove that distance correlates with loss difference.

- Figure 1 is suggestive, but the paper does not include full summary statistics. It is hard to tell how consistent the variance reduction is across cluster sizes or training checkpoints.

- The use of balanced K-means may inflate the observed variance reduction. There is no direct comparison to random or unbalanced clustering, or to distance-based correlation.

- Section 3 is hard to follow. The experimental setup is long, and results are sometimes buried. It would help to clearly state the goal and takeaway for each experiment.

- The comparison between off-the-shelf and specialized embeddings feels a bit unfair. The specialized embeddings are pulled from models trained on the same data. That likely explains some of the improvement.

- The efficiency argument is not convincing. While some of the embeddings require no forward pass, they still rely on a trained model. That training cost is not addressed.

---

> ### Author Rebuttal · Authors · 2025-07-31
>
> We thank the reviewer for their detailed feedback! We appreciate that you find that our method addresses an "interesting gap in the literature" and is "well-suited" to the pretraining context. We respond to individual comments below:
>
> > **"The strongest claims rely on the idea that low embedding distances mean similar pretraining loss, but the actual evaluation uses variance within clusters, not pairwise distance. This does not prove that distance correlates with loss difference."**
>
> We would like to clarify the setup for our variance reduction experiments. Our goal is precisely to quantify what the reviewer highlights — whether examples that are close in embedding space also exhibit similar pretraining loss.
>
> Since we cluster in the embedding space, each cluster now contains groups of examples that are all similar to each other. Then, for every cluster, we measure the training loss for all points in the cluster and take the variance of these losses  to quantify the similarity in the losses of these points that are also similar to each other in the embedding space.
>
> Of course this is for a single cluster, and so we report the average of this variance across all clusters, normalized by the overall loss variance in the dataset, which is our variance reduction ratio.
>
> We would also like to note that we can only reliably test similarity in this one direction (as our approach does) --- that similar examples have similar training loss. The **converse is not necessarily true** --- two very different examples can have the same loss because they happen to be equally "easy" or equally "hard". This **asymmetry** is an important reason we chose our cluster-based approach rather than pairwise correlation. We will highlight this more clearly in our revision.
>
> > **"Figure 1 is suggestive, but the paper does not include full summary statistics. It is hard to tell how consistent the variance reduction is across cluster sizes or training checkpoints."; "complete results for the variance reduction analysis, such as how it changes across different cluster sizes, seeds, or pretraining checkpoints?"**
>
> We agree that Figure 1 serves primarily as a qualitative illustration. However, Figures 2 and 3 present **exactly these requested summary statistics** — showing variance reduction trends across different **cluster sizes** (Figure 2) and **pretraining checkpoints** (Figure 3).
>
> We find the relative performance of different embeddings is consistent across both axes. This supports the robustness of our conclusions.
>
> > **"The use of balanced K-means may inflate the observed variance reduction. There is no direct comparison to random or unbalanced clustering... Why did you choose balanced K-means for clustering?"**
>
> We chose balanced K-means to ensure a fair and stable comparison across different embedding models. Without balancing, clustering can yield singleton clusters or overly large groups, leading to unstable variance estimates and increased noise. Balanced clustering improves comparability across embedding models with differing geometric properties. A random clustering would serve as a lower bound, achieving variance reduction ≈ 1 by design (which is far worse than the considered embeddings' clusters).
>
> Overall, this choice of balanced clustering does not inflate the results in a biased way, as all methods are evaluated under the same balanced clustering scheme, making their relative performance directly comparable.
>
> > **"Section 3 is hard to follow. The experimental setup is long, and results are sometimes buried. It would help to clearly state the goal and takeaway for each experiment."**
>
> Thank you for the helpful suggestion. We will revise Section 3 to clearly present the goal and takeaway for each evaluation metric to improve clarity and structure.
>
> > **"The comparison between off-the-shelf and specialized embeddings feels a bit unfair. The specialized embeddings are pulled from models trained on the same data. That likely explains some of the improvement. How do you address the potential circularity in using embeddings from models trained on the same data used in the evaluation? Would these embeddings generalize to new domains or corpora where no model has yet been trained?"**
>
> This is a fair point — and one we agree with. In fact, one of our core arguments is that embedding models should ideally be trained on the same data used for pretraining. A key finding in our paper is that even small proxy models trained on the same corpus (e.g., 200M parameters) can outperform large off-the-shelf models. This is not standard practice today (see [1,2]), and we argue that it should be.
>
> The notion of generalizing to new domains is somewhat orthogonal to our main use case, as we could always use such data to train a new embedding model. And since the embedding models are small and much faster to train, you could simply train this embedding model when you obtain data from new domains.
>
> > **"The efficiency argument is not convincing. While some of the embeddings require no forward pass, they still rely on a trained model. That training cost is not addressed."**
>
> We agree that a trained model is required, but the associated cost of training that model is modest. The model is small (200M parameters) and trained on only a 20% slice of the full dataset. Compared to training a 1.7B model on 200B tokens, this is a negligible fixed cost. Moreover, this cost is a one-time investment, which is amortized across all embedding use cases, and this slight marginal cost is easily offset by the significant improvements in data curation quality. We will clarify this point in our revision.
>
> Thank you again for your detailed review. We hope these clarifications address your concerns and help reinforce the value of our contributions.
>
> [1] Vo, et. al. Automatic Data Curation for Self-Supervised Learning: A Clustering-Based Approach.
>
> [2] Chang, et. al. Data Curation Alone Can Stabilize In-context Learning.

---

> > ### Comment · Reviewer_VPjN · 2025-08-06
> > **Response to Comment**
> >
> > Thank you for the detailed response. However, after reading the rebuttals and other reviews, my main concerns are not resolved. The paper's central claim that embedding distance captures training dynamics is not convincingly demonstrated. The use of variance within clusters does not directly support this claim, and more targeted analysis is needed. I also still find the comparisons problematic, since the embeddings are taken from models trained on the same data. This setup limits generality and makes the improvements less surprising, even if the authors state the current standard is different. I believe my assessment is fair, and the paper is not quite ready for publication.

---

### Official Review · Reviewer_qKM8 · 2025-07-03

**Clarity:** 3
**Significance:** 3
**Originality:** 3
**Rating:** 5
**Confidence:** 4

**Summary:**

The paper introduces an evaluation framework for deciding whether a text-embedding model is suitable for curating pre-training data for large language models. The authors motivate and propose three metrics that help to evaluate the quality of the embedding model. They also pointed out that the LM Token Embeds (require no forward pass), and LM Output Embeds outperform the off-the-shelf embedding models with repect to the three metrics.

The three metrics are variance-reduction ratio, downstream score of the model on a selected subset, and cluster-purity. The three metrics are simple and intuitive. The paper provides an evaluation framework for the community.

The paper is well-written and because using LM Token Embeds and LM Output Embeds is not a general practice, the paper is a good contribution to the community.

**Questions:**

1. Have the authors provided detailed metrics—including total FLOPs, elapsed time, for the 200 M-parameter proxy model, the RAC clustering stage, and the full 1.7 B pre-training run?

2. Do the authors think the three metrics still hold and are useful when considering a corpus that has large portion of code?

3. How sensitive is the the downstream performance to the $\epsilon$ grid? In line 175, I see that we would like to select the possible largest $\epsilon$.  It would be helpful to know whether the final results are robust to this setting; if performance varies little with $\epsilon$, the grid search could be narrowed and further compute saved.

**Ethical Concerns:**

["NO or VERY MINOR ethics concerns only"]

**Final Justification:**

The authors have provided the runtime of proxy model training, RAC clustering and 1.7B model pretaining to answer my question. Hope to see more on the robustness of $\epsilon$ can be exploited to speed up the process, as currently the clustering cost 1-2 days.

**Limitations:**

yes

**Paper Formatting Concerns:**

No formatting concerns.

**Quality:**

4

**Strengths And Weaknesses:**

The paper is well-written and the authors have done a good job of motivating the three metrics, providing a clear definition of the metrics. The authors also carried out large-scale pretraining experiments (1.7B language models with 512 v5 TPUs) to evaluate the three metrics. Pre-training experiments are expensive to carry out, thus the contribution and insights are valuable. The paper has good clarity, the writing is clear and the figures are well-designed. Line 128 has a small grammar error. In terms of originality, since both LM Token Embeds and LM Output Embeds are not general practice, the paper is a good contribution to the community. The three metrics are simple intuitive, and easy to carry out. For weakness, only one corpus (the Pile) and one model architecture were evaluated. However, as noted by the authors the conclusion stands alone without the evaluation of other corpora. Another potential weakness for originality is the statistical tool, variance ratios, and cluster purity are classical, but the application is new.

---

> ### Author Rebuttal · Authors · 2025-07-31
>
> We thank the reviewer for their positive assessment and for finding our work to be a "good contribution to the community" with "valuable" insights. We are glad you found the paper well-written and the experiments insightful. We address your questions below.
>
> > **"Have the authors provided detailed metrics—including total FLOPs, elapsed time, for the 200 M-parameter proxy model, the RAC clustering stage, and the full 1.7 B pre-training run?"**
>
> We will add these details to the Appendix in our revision for completeness. To summarize here: while we did not log the exact total FLOPs, the approximate run times were as follows:
> * **200M Proxy Model Training**: Took less than 1 day on 64 v5 TPUs.
> * **RAC Clustering**: Took approximately 1-2 days to run on the full dataset.
> * **1.7B Model Pretraining**: Each run took approximately 3 days on 512 v5 TPUs.
>
> > **"Do the authors think the three metrics still hold and are useful when considering a corpus that has large portion of code?"**
>
> Yes, we believe the framework is directly applicable. We would like to remark that one data source for the Pile is Github, so this pretraining corpus already contains a substantial amount of code. Furthermore, when computing variance reduction, these are clusters produced by a balanced K-means clustering algorithm, which means that even with large amounts of code, this slice of the dataset will be partitioned into many smaller clusters. This allows for a granular and meaningful analysis of similarity within the code domain, not just between code and other text types.
>
> > **"How sensitive is the downstream performance to the $\epsilon$ grid? In line 175, I see that we would like to select the possible largest. It would be helpful to know whether the final results are robust to this setting; if performance varies little with $\epsilon$, the grid search could be narrowed and further compute saved."**
>
> This is an excellent question regarding robustness. The grid search for $\epsilon$ is only used to find a threshold that produces a sufficiently fine-grained clustering (i.e., a target number of clusters) for our data selection budget. Crucially, we did not optimize this hyperparameter for each embedding model; the same process was used for all of them to ensure a fair comparison. We believe that performance is likely robust to small changes in ϵ as long as it yields a comparable number of clusters. We will clarify this in the revision and note that narrowing the grid is indeed a promising efficiency improvement if one is not comparing across embedding methods.

---

> > ### Comment · Reviewer_qKM8 · 2025-08-06
> >
> > Thank you for providing the runtime of proxy model training, RAC clustering and 1.7B model pretaining to answer my question. Hope to see more on the robustness of can be exploited to speed up the process, as currently the clustering cost 1-2 days. The clarifications have addressed all of my concerns, and I will maintain my recommendation for acceptance.

---

> > > ### Author Response · Authors · 2025-08-07
> > >
> > > Thank you again for your review. We appreciate your efforts during the review process and your positive recommendation of our work!

---

### Official Review · Reviewer_CAUd · 2025-07-04

**Clarity:** 2
**Significance:** 2
**Originality:** 2
**Rating:** 4
**Confidence:** 2

**Summary:**

The authors present a new evaluation framework for the understudied area of similarity metrics used in language model pretraining data curation. Using their framework, they show that off-the-shelf embeddings—despite their widespread use—often underperform even simple, specialized alternatives such as an average of token embeddings that requires no forward pass.

**Questions:**

1. It would be interesting to explore whether representations from relatively smaller general-purpose models (e.g., around 1.7B parameters) could serve as effective signals for data selection. With the aid of acceleration techniques, the inference cost of such models is unlikely to be a limiting factor.
2. The data similarity metric proposed in this work appears to primarily capture textual style similarity. However, for modern large models, the actual content of the text is equally critical. Could relying on representation-space similarity as the selection criterion inherently introduce biases on certain datasets?

**Ethical Concerns:**

["NO or VERY MINOR ethics concerns only"]

**Final Justification:**

This paper tackles a simple yet fundamental and underexplored problem—pre-training data selection—and proposes a practical metric evaluation protocol with clear applicability. Despite presentation issues and limited discussion on redundancy reduction, I think the strengths of this paper outweigh its weaknesses.

**Limitations:**

Yes, the authors have discussed the limitations.

**Quality:**

2

**Strengths And Weaknesses:**

### Strength

1. This paper focuses on a simple and fundamental, yet underexplored problem: the selection and curation of pre-training data. Under the current data-driven paradigm, this may offer some meaningful insights.
2. The authors propose a practical metric evaluation protocol to assess the suitability of the data selection criteria before applying them in actual data selection.

### Weakness

1. The presentation of the paper could be further improved. It is recommended to include a flowchart or schematic diagram to summarize the proposed data similarity metric evaluation protocol.
2. The paper could further elaborate on the impact of redundancy reduction in the context of modern language model pre-training, which would better highlight the significance of this work.

---

> ### Author Rebuttal · Authors · 2025-07-31
>
> We thank the reviewer for their thoughtful feedback and are encouraged that you view our work as addressing an “understudied area” in how notions of similarity are used in data curation for language models. We address your individual comments below:
>
> > **"It is recommended to include a flowchart or schematic diagram to summarize the proposed data similarity metric evaluation protocol"**
>
> Thank you for this feedback! Our main evaluation protocol is described in Subsections 3.1, 3.2, and 3.3, and we plan to include a figure that visualizes these components in our camera-ready revision.
>
> > **"The paper could further elaborate on the impact of redundancy reduction in the context of modern language model pre-training, which would better highlight the significance of this work"**
>
> We believe that redundancy reduction (and deduplication) plays an important role in modern language model pretraining. Prior work has shown that exact deduplication can lead to significant performance improvements [1], and that semantic redundancy reduction can result in more efficient pretraining [2]. Within this broader context, our work analyzes **how similarity should be measured** — a key but **underexplored** design choice in these prior works on redundancy reduction. We will add this clarification to our revision.
>
> > **"It would be interesting to explore whether representations from relatively smaller general-purpose models (e.g., around 1.7B parameters) could serve as effective signals for data selection. With the aid of acceleration techniques, the inference cost of such models is unlikely to be a limiting factor."**
>
> This is an excellent point. We use a 200M parameter model as a computationally efficient proxy to generate specialized embeddings, and our results show that even this smaller, specialized model outperforms larger, off-the-shelf alternatives. We agree with the reviewer that using a 1.7B parameter model that has been pretrained on the same corpus would be effective and likely yield even stronger results, which would further reinforce one of our central claims that embeddings specialized to the pretraining corpus are highly performant in measuring similarity. We believe that scaling up such embedding models is an interesting direction for future work, and we will add this discussion to our revision.
>
> > **"The data similarity metric proposed in this work appears to primarily capture textual style similarity. However, for modern large models, the actual content of the text is equally critical. Could relying on representation-space similarity as the selection criterion inherently introduce biases on certain datasets?"**
>
> This is a great question. Our qualitative analysis in Figure 1 suggests that the learned representations capture **more than just superficial textual style**. For instance, two of the examples clustered together have **related topics** (e.g., scientific or historical content) despite significant differences in their specific formats. This indicates that the learned representations are sensitive to content semantics, not just stylistic features.
>
> That said, we certainly acknowledge that the potential inductive biases of any given representation space are a complex issue, even when they are learned in a general-purpose manner (e.g., through standard pretraining objectives such as next-token prediction). We will add a note to our discussion clarifying that while our method shows strong empirical results, a deeper analysis of these biases is an important direction for future research.
>
> Thanks again for your thorough review! We hope our responses and clarifications have fully addressed your concerns.
>
> [1] Tirumala, et. al. D4: Improving llm pretraining via document de-duplication and diversification.
>
> [2] Abbas, et. al. Semdedup: Data efficient learning at web-scale through semantic deduplication.

---

> > ### Comment · Reviewer_CAUd · 2025-08-07
> > **Thanks for your response!**
> >
> > Sorry for the late rely. I appreciate the detailed response by the authors. After reading the rebuttals and other reviews, I still share similar perspectives of Reviewer Q6cD and VPjN. The main concern is the motivation and mechanism behind data similarity-based metric are not properly presented. Considering that the authors trained the metric on the same data, could this data selection be integrated into the pre-training stages or as some indicators to enhance the data quality? And why the proposed similarity works should be further analyzed. I understand that more experiments are impossible at this time and more explanations are enough for me. I will also read the responses to other reviewers.

---

> ### Author Response · Authors · 2025-08-07
>
> We appreciate your continued engagement!
>
> ---
>
> > "The main concern is the motivation and mechanism behind data similarity-based metric are not properly presented."
>
> Thank you for your feedback --- we hope to clarify our motivations behind our proposed metrics below, and we have also incorporated this into our revision:
>
> * **Variance Reduction**: The primary goal here is to find an efficient proxy for how "difficult" the model finds an example. An embedding space where nearby points have similar training losses is highly valuable. It means we can group examples by difficulty, which is a key requirement for advanced sampling strategies like curriculum learning or importance resampling, without needing to compute the loss for every single example.
> * **Pretraining efficiency**: This is the ultimate, end-to-end test of an embedding's utility. We directly measure whether using a given similarity metric to de-duplicate and diversify a dataset translates to real-world gains on downstream tasks. This grounds our analysis in the final goal of pre-training: creating a powerful and generalizable language model.
> * **Data source purity**: This metric serves as a crucial check for interpretability and trustworthiness. By measuring if embeddings can naturally separate human-curated data sources (e.g., Wikipedia vs. GitHub), we verify that they are capturing meaningful, high-level structures in the data. This is essential for building specialized datasets or controlling data mixtures with confidence. Controlling for data sources has been shown to be useful for improving performance [1].
>
> [1] Wettig, et. al. Organize the Web: Constructing Domains Enhances Pre-Training Data Curation.
>
> ---
>
> > "Considering that the authors trained the metric on the same data, could this data selection be integrated into the pre-training stages or as some indicators to enhance the data quality?"
>
> We'd like to highlight that we have already done this precisely in **Table 1**, where we have incorporated the trained similarity metrics (as well as off-the-shelf models) in a full pretraining run, using the diversity-based data selection process outlined in Section 3.2. This precisely gives the indicator of improved data quality that has been integrated into pretraining that the reviewer has requested. Our results show that this leads to improvements on a large number of downstream tasks.
>
> ---
>
> > "why the proposed similarity works should be further analyzed"
>
> The choice of using an embedding model for measuring similarity (as done in many different works [2,3,4,5]) is a **fundamental design choice** in the data curation pipeline for pretraining. As such, picking exactly which embedding model leads to (1) strongest performance and with (2) lowest computational cost is crucial, given that these data selection algorithms require running inference of these embedding models on trillions of tokens.
>
> Analyzing this is challenging due to the high cost of pre-training (e.g., each run taking $20000 in Cloud TPU costs), which is precisely why our scalable and efficient evaluation framework is so crucial. It provides a reliable way to validate this mechanism and select the best embedding model before committing to a full, costly pre-training run.
>
> [2] Abbas, et. al. Effective pruning of web-scale datasets based on the complexity of
> concept clusters.
>
> [3] Abbas, et. al. Semdedup: efficient learning at web-scale through semantic deduplication.
>
> [4] Tirumala, et. al. D4: Improving llm pretraining via document de-duplication and diversification.
>
> [5] Vo, et. al. Automatic Data Curation for Self-Supervised Learning: A Clustering-Based Approach.
>
> ---
>
> Thank you again for your review! *We also hope that you consider updating the overall evaluation score for our work if we have addressed most of your concerns*.

---

### Decision · Program_Chairs · 2025-09-17

**Decision:**

Accept (poster)

**Comment:**

1. **Summary**
   The paper introduces a framework for evaluating similarity metrics used in pretraining data curation for large language models. The framework consists of three criteria: (a) correlation between embedding distance and pretraining loss, (b) downstream performance when embeddings guide diversity-based data curation, and (c) cluster purity with respect to human-curated data sources. Experiments use The Pile and a 1.7B parameter UL2 decoder-only model trained on 200B tokens. Results show that off-the-shelf embeddings (e.g., USE, BERT, Gecko) underperform compared to simple embeddings derived from models trained on the same corpus (LM Token Embeds, LM Output Embeds). The work highlights that task-aligned and domain-specialized embeddings are more effective for data curation than generic ones.

2. **Strengths**

   * Addresses an underexplored but important problem: evaluation of similarity metrics for pretraining data curation.
   * Provides a clear, three-part evaluation framework grounded in pretraining dynamics rather than retrieval benchmarks.
   * Experimental validation is large-scale (1.7B model, 200B tokens), lending weight to conclusions despite computational challenges.
   * Results are surprising and practically relevant: trivial embeddings (averaged input embeddings) often outperform sophisticated off-the-shelf models.
   * Strong clarity of motivation: moves the field toward more principled embedding evaluation for data selection.

3. **Weaknesses**

   * Gains on downstream tasks are modest (about 2% on average), which several reviewers flagged as insufficient to justify strong claims (Q6cD, VPjN).
   * Loss-correlation and cluster-purity evaluations are not directly tied to downstream improvements, and the authors did not provide new evidence during rebuttal that these metrics translate to stronger downstream performance (Q6cD).
   * Evaluation restricted to The Pile and a single model size, raising questions of generality that remain unaddressed (qKM8, VPjN).
   * Concerns about circularity: embeddings trained on the same data naturally outperform off-the-shelf models, which undermines the claim of broad applicability (VPjN). The rebuttal acknowledged this but insisted it should be standard practice, which did not fully resolve the critique.
   * Methodological clarity issues remain: variance reduction as a proxy for distance-loss correlation was contested (VPjN), balanced K-means potentially inflates results (VPjN), and although the authors defended these choices, no additional quantitative validation was provided
   * Presentation issues (dense experimental section, limited clarity on clustering choices and robustness). The authors promised revisions but no evidence could be evaluated at review time

4. **Decision Rationale**

   While the framework is well-motivated, the empirical evidence does not convincingly demonstrate that embedding distance reliably captures pretraining dynamics, nor that the proposed evaluations generalize beyond the tested setting. Critical reviewer concerns about robustness, generality, and modest impact were not fully addressed in the rebuttal, and at least one reviewer maintained a rejection on these ground.

5. **Reviewer Discussion and Rebuttal Dynamics**

   * Reviewer CAUd initially raised concerns about presentation and redundancy reduction. The authors promised schematic figures and clarified the role of redundancy reduction and the impact of using smaller models for embeddings. This response was satisfactory.
   * Reviewer qKM8 requested runtime and FLOP details, robustness to clustering thresholds, and applicability to code-heavy corpora. The authors provided detailed runtimes and clarified robustness and applicability. The reviewer maintained an acceptance.
   * Reviewer VPjN raised stronger concerns regarding variance reduction as an indirect proxy for distance-loss correlation, potential inflation from balanced K-means, and limited generality of in-domain embeddings. The authors clarified their methodology, emphasized comparability across embeddings, and argued that training small proxy models is inexpensive. Nevertheless, this reviewer maintained a borderline reject.
   * Reviewer Q6cD considered the methodology rigorous and original but noted modest improvements and limited linkage of metrics to downstream performance. The authors reframed results in terms of data efficiency and clarified how loss-correlation and cluster-purity could inform sampling strategies. The reviewer kept a borderline accept.

Given the balance of reviews, the critical weaknesses, and the lack of resolution during the rebuttal period, I recommend rejection.